

# A retrospective study of immunoglobulin E as a biomarker for the diagnosis of acute ischemic stroke with carotid atherosclerotic plaques

Wenwen Jiang[1], Jindou Niu[2], Hongwei Gao[1], Yingqiang Dang[1], Meijiao Qi[1] and Yumei Liu[1]

[1] Laboratory Medicine Center, Lanzhou University Second Hospital, Lanzhou, Gansu Province, China
[2] Department of Clinical Laboratory, Maternal and Child Health Hospital of Gansu Province, Lanzhou, Gansu Province, China

Corresponding author
Yumei Liu, 13919833218@163.com

## ABSTRACT

**Objective.** In this study, serum markers of acute ischemic stroke (AICS) with carotid artery plaque were retrospectively evaluated to establish a basis for discovering serological indicators for early warning of acute ischemic stroke (AICS).

**Methods.** A total of 248 patients with AICS were enrolled in Lanzhou University Second Hospital from January 2019 to December 2020. The study population included 136 males and 112 females, $64 \pm 11$ years of age. Of these, there were 90 patients with a transient ischemic attack (TIA), including 60 males and 30 females, aged $64 \pm 8$ years old. Patients with AICS were stratified by carotid ultrasound into a plaque group ($n = 154$) and a non-plaque group ($n = 94$). A total of 160 healthy subjects were selected as the control group. Serum lipoprotein-associated phospholipase A2 (Lp-PLA2), amyloid A (SAA), immunoglobulin E (IgE), D-dimer (D-D), total cholesterol (TC), triglyceride (TG), and low-density lipoprotein cholesterol (LDL-C) were collected from all subjects. Multivariate logistic regression was used to analyze the risk factors of AICS with carotid plaque. ROC curve was used to analyze the diagnostic efficacy of AICS with carotid plaque.

**Results.** The IgE, Lp-PLA2, SAA, LDL-C, TC, TG, and D-D levels in the AICS group were higher than those in the TIA group and healthy control group ($P < 0.05$). The IgE level was significantly higher than that in the healthy control group and TIA group. The IgE level in the AICS plaque group was significantly higher than that in the AICS non-plaque group ($P < 0.01$), and the Lp-PLA2 level was also different ($P < 0.05$). The incidence of AICS was positively correlated with Lp-PLA2, TC, IgE, TG, D-D, SAA and LDL-C ($r = 0.611, 0.499, 0.478, 0.431, 0.386, 0.332, 0.280$, all $P < 0.05$). The incidence of AICS with plaque was only positively correlated with IgE and Lp-PLA2 ($r = 0.588, 0.246, P < 0.05$). Logistic regression analysis showed that IgE and Lp-PLA2 were independent risk factors for predicting the occurrence of AICS with carotid plaque ($P < 0.05$). ROC curve analysis showed that the AUC of IgE (0.849) was significantly higher than other indicators; its sensitivity and specificity were also the highest, indicating that IgE can improve the diagnostic efficiency of AICS with carotid plaque.

**Conclusion**. IgE is a serum laboratory indicator used to diagnose AICS disease with carotid plaque, which lays a foundation for further research on potential early warning indicators of AICS disease.

# INTRODUCTION

Stroke is a common cerebrovascular disease with a rapid onset, high disability rate, and a high fatality rate. The incidence types include ischemic stroke and hemorrhagic stroke. At present, acute ischemic stroke (AICS) accounts for about 70%–80% of strokes globally (*Wang et al., 2016*; *Barthels & Das, 2020*). In China, the incidence of a first stroke for those between the ages of 40 to 74 increases by 8.3% annually, on average, and more than 1.9 million people die from stroke every year. The prevention and treatment of stokes remains challenging (*Chinese Stroke Prevention and Treatment Report 2019, 2020*).

At present, a stroke diagnosis is based on clinical symptoms and their onset and relies on a clinical evaluation and imaging diagnosis (*Dong et al., 2017*). However, the current diagnosing method risks poor prognosis or disability for the patient. Serological indexes have the advantages of being simple to operate with a strong repeatability. Serum biomarkers found in the early stages of the disease may act as an early warning of the occurrence of AICS and can be found during a positive imaging examination, or even before the appearance of symptoms and imaging changes. The biomarkers may assist in early intervention, such as preventive treatment, lifestyle change, *etc.*, and may effectively prevent the occurrence of AICS disease.

In a clinical evaluation of normal blood lipids, the most commonly used serum biomarkers are total cholesterol (TC), triglyceride (TG), and low-density lipoprotein cholesterol (LDL-C). However, these indicators are abnormal in many diseases, such as diabetes, obesity, genetics, *etc.*, and cannot provide a good indication of AICS. In particular, there is no warning of AICS with carotid plaques.

In recent years, IgE has been studied in greater detail in the cardiovascular and cerebrovascular system. Studies have shown that the body's allergic reaction plays an important role in the pathogenesis of atherosclerosis. Immunoglobulin E (IgE) is synthesized and released by B lymphocytes (*Pate et al., 2010*) and is a key component protein of allergen/antigen signal transduction response, participating in atopic and systemic allergic reactions (*Yang, Zhang & Cao, 2020*). It can directly lead to local inflammatory response and promote the occurrence and development of atherosclerosis by directly acting on endothelial cells and smooth muscle cells (*Wang et al., 2018*).

Atherosclerosis is the result of a series of chronic inflammatory reactions, and inflammation runs through the pathophysiological process of the occurrence, development and outcome of atherosclerosis. Lipoprotein-associated phospholipase A2 (Lp-PLA2) is a novel calcium-independent lipolysis enzyme, also known as platelet activator

acetylhydrolase, which binds to circulating low density lipoprotein cholesterol and promotes the initiation of atherosclerosis. As a new marker of the vascular inflammatory response, it plays an important role in the formation and development of plaque (*Zhang, Lu & Zhang, 2018*), and can affect the stability of atherosclerotic plaque, promote plaque rupture, and ultimately lead to acute cerebrovascular events (*Bonnefont-Rousselot, 2016*; *Qin et al., 2020*).

Serum amyloid A (SAA) is an acute reactive protein synthesized by the liver, which is an auxiliary diagnostic tool for acute pancreatitis. Recent studies have found that SAA can serve as a marker of an acute inflammatory response. It can mediate the functional transformation of high-density lipoprotein, affect cholesterol transport, promote the chemotaxis and adhesion of monocytes/macrophages, activate multiple inflammatory signaling pathways, promote atherosclerosis and thrombosis (*Abouelasrar Salama et al., 2021*), and participate in the formation of atherosclerotic plaque and the progression of ischemic stroke (*Dinler et al., 2020*). D-dimer (D-D) is an important marker of fibrinolytic and coagulation system activation, and studies have shown that D-D can be used to evaluate the progression of stroke disease (*Zhang et al., 2019*).

Based on the above research, this study focuses on the serum or plasma indicators associated with atherosclerosis in carotid plaques of AICS patients, non carotid plaque of AICS patients, TIA patients and healthy controls. This study analyzed the combination of carotid plaques in patients with acute ischemic stroke and the high diagnostic value of laboratory indicators. It lays a foundation for further establishing the laboratory index of early warning AICS.

## MATERIALS & METHODS
### Objects
This study was a retrospective cohort study approved by the Ethics Committee of the Second Hospital of Lanzhou University (2021A-001). Participants provided informed consent in the form of an on-site confirmation and signatures. Two hundred forty-eight patients with acute ischemic stroke (AICS), including 136 males and 112 females aged $64 \pm 10$, were enrolled in the Second Hospital of Lanzhou University from January 2019 to December 2020. Ninety patients with a transient ischemic attack (TIA) were selected, including 60 males and 30 females, aged $64 \pm 8$ years.

#### Inclusion criteria of AICS patients
All patients met the diagnostic criteria of acute ischemic stroke (acute cerebral infarction) in the *The Neurology Branch of the Chinese Medical Association & Cerebrovascular Disease Group of Chinese Medical Association (2015)*: (1) acute onset; (2) neurological deficits in local lesions (weakness or numbness of one side of the face or limb, language impairment, *etc.*), or comprehensive neurological defects; (3) the duration of symptoms or signs is not limited (when imaging shows responsible ischemic lesions), or is greater than 24 h (when imaging shows responsible lesions); (4) exclude non-vascular causes; (5) brain CT/MRI excluded cerebral hemorrhage; (6) no anticoagulant drugs or lipid-lowering drugs were taken before onset.

### Exclusion criteria for AICS patients

(1) Brain lesions caused by traumatic brain injury, poisoning, post epileptic state, tumor stroke, hypertensive encephalopathy, abnormal blood glucose, encephalitis, and severe dysfunction of vital organs; (2) hemorrhagic stroke; (3) transient ischemic attack (TIA); (4) acute ischemic stroke complicated with tumor and autoimmune diseases; (5) severe infection and heart, liver and kidney diseases; (6) pregnancy, lactation or long-term use of contraceptives; (7) taking hypolipidemic and anticoagulant drugs; (8) poor compliance.

### Inclusion criteria for TIA patients

(1) Transient neurological deficits caused by local cerebral or retinal ischemia, with typical clinical symptoms lasting no more than 1 h, and (2) no evidence of acute cerebral infarction on imaging.

### Grouping

(1) Acute ischemic stroke group (AICS) group ($n = 248$), carotid artery ultrasound examination stratified into plaque group ($n = 154$) and non-plaque group ($n = 94$); (2) transient ischemic attack group (TIA) ($n = 90$); (3) healthy control group ($n = 160$). The healthy control group was composed of 160 healthy subjects who underwent a physical examination at the health management center of our hospital during the same period were included in the control group, including 102 males and 58 females, aged $63 \pm 6$ years old, who did not meet the above diagnostic criteria of AICS and TIA. The brain CT/MRI imaging examinations of this group were negative.

### Criteria for the definition of carotid plaque

The criteria were defined according to the American Ultrasound Conference in 2003 and the Chinese Ultrasound Diagnostic Guidelines in 2009 (*Huang & Yao, 2019*). The IMT thickness was measured at the distal end of the common carotid artery 1.5~2.0 cm away from the bifurcation. An IMT $\geq$ 1.0 mm was considered thickening, and an IMT $\geq$ 1.5 mm was considered as plaque formation. During plaque formation, the degree of carotid artery stenosis should be determined. The degree of stenosis was divided into mild stenosis (<50%), moderate stenosis (50–69%), severe stenosis (70–99%), and complete occlusion (plaque was completely filled with lumen without blood flow). AICS patients meeting the above criteria were assigned to the AICS plaque group. Patients with AICS without these ultrasound indications were grouped in the AICS non-plaque group.

## Methods

### General data

The general information for each patient, including age, sex, the history of smoking, hypertension and diabetes were recorded in detail.

### Laboratory indicators

A total of 4 mL of peripheral venous blood was collected from the AICS patients and TIA patients 12 h after admission and was used to detect serum markers. Meanwhile, 3 mL peripheral venous blood was collected from a blue vacuum blood vessel tube containing sodium citrate (1:9) for the detection of D-dimer. In the healthy control group, 4 mL

peripheral venous blood was collected in the morning, after fasting, to detect serum markers in an ordinary vacuum blood collection tube without an anticoagulant. A total of 3 mL peripheral venous blood was collected in a blue vacuum blood collection tube containing sodium citrate (1:9) to detect D-dimer. After the collection vessel was filled with blood, it rested for 0.5 h; the blood was then centrifuged at 4,000 r/min for 10 min to separate the blood and plasma. The serum was used to detect lipoprotein-associated phospholipase A2 (LP-PLA2), amyloid A (SAA), immunoglobulin E (IgE), total cholesterol (TC), triglyceride (TG), and low-density lipoprotein cholesterol (LDL-C), and plasma was used to detect D-dimer (D-D).

Lp-PLA2, SAA, IgE, TC, TG, and LDL-C were detected by a German Roche Cobas 8000 automatic biochemical analyzer (Roche Diagnostics, Mannheim, Germany). The Lp-PLA2 and IgE kits were provided by Desai Diagnostic Systems Co. Ltd. (Shanghai, China). Continuous monitoring was used to identify Lp-PLA2; the particle enhanced immune transmission turbidimetric resistant method was used to detect IgE. Ningbo Purebio Biotechnology Co., Ltd (Zhe Jiang, China) provided the SAA kit and latex enhanced immune turbidimetry was used as the detection method. The TC, TG, and LDL-C detection kits were Roche kits. TG were detected using the enzyme method. TC were detected using cholesterol oxidase, and LDL-C was detected using enzyme colorimetry. D-dimer (D-D) was detected using the SYSMEX CS5100 hemagglutination analyzer, and equipped with the Siemens (Ade-Behring) dealing hemagglutination reagent. Immunoturbidimetry was used as the detection method. The testing materials were required to pass performance verification, calibration, and indoor quality control tests prior to testing the specimens. Serum or plasma samples were tested according to the instrument operating procedures from the manufacturers to ensure the accuracy and comparability of results.

## Statistical analysis

SPSS 25.0 statistical software was used for the statistical analysis of the data. The $X \pm S$ LSD-test expressed the standard distribution measurement data of age and laboratory indicators and was used to compare between two groups. One-way ANOVA was used to compare multiple groups. Non-normal distribution data of age and laboratory indicators were expressed as M (Q1, Q3) and defined. The Mann–Whitney U rank-sum test was used for inter-group differences. The comparable enumeration data of count data were described by frequency, and $X^2$ test was used to compare multiple groups. The correlation between AICS and its indicators was analyzed by bivariate Spearman rank correlation analysis. Logistic regression was used to analyze the independent risk factors of AICS in patients with carotid plaques. A patient's AICS status was taken as the dependent variable (yes = 1, no = 0), and statistically significant and clinically significant variables in univariate analysis were taken as independent variables to analyze the potential influencing factors for the occurrence of AICS. The diagnostic value of IgE and other indicators for AICS with carotid plaque was investigated by the ROC curve and the area under the ROC curve (AUC) of each hand. The sensitivity was calculated, and the specificity and diagnostic efficacy of the different indicators were compared. The AUC was compared by Z test and bilateral tests test level $a = 0.05$.

**Table 1** Comparison of test results of AICS group, TIA group and healthy control group.

| Index | Healthy control group (n = 160) | TIA group (n = 90) | AICS group (n = 248) | $F/Z/X^2$ value | p value |
|---|---|---|---|---|---|
| Gender (e.g., male/female) | 102/58 | 60/30 | 136/112 | 5.344 | 0.069 |
| Smoking (e.g., yes/no) | 53/107 | 31/59 | 80/168 | 0.147 | 0.929 |
| DM (e.g., yes/no) | 17/143 | 8/82 | 40/208 | 4.274 | 0.118 |
| HTN (e.g., yes/no) | 11/149 | 10/80 | 31/217 | 3.343 | 0.188 |
| Age (years,x ± s) | 63 ± 6 | 64 ± 8 | 64 ± 11 | 0.553 | 0.576 |
| LDL-C (mmol/L,M(Q1,Q3)) | 1.69(1.53,2.18)[b] | 2.01(1.64,2.22)[a] | 2.21(1.68,2.82)[ab] | 34.530 | 0.000 |
| TC (mmol/L,M(Q1,Q3)) | 2.77(2.53,2.99) | 2.77(2.57,3.00) | 3.62(2.98,4.16)[ab] | 119.821 | 0.000 |
| TG (mmol/L,M(Q1,Q3)) | 1.06 (0.80,1.25) | 1.07(0.74,1.27) | 1.52(1.09,1.99)[ab] | 91.684 | 0.000 |
| IgE (U/ml,M(Q1,Q3)) | 4.77(2.55,9.65) | 8.30(0.00,23.20) | 35.85 (8.25,76.27)[ab] | 105.144 | 0.000 |
| Lp-PLA2 (U/L, M(Q1,Q3) ) | 247.00 (215.00,274.5) | 257.5(178.25,333.00) | 375.00(298.00,472.25)[ab] | 175.442 | 0.000 |
| SAA (mg/L,M(Q1,Q3)) | 3.00 (2.00,4.00) | 3.50(2.00,4.25) | 4.00 (2.00,13.25)[ab] | 50.669 | 0.000 |
| D-D (mg/L,M(Q1,Q3)) | 0.22(0.14,0.37)[b] | 0.38(0.18,0.54)[a] | 0.48 (0.21,1.20)[ab] | 10.175 | 0.001 |

**Notes.**

Compared with healthy control group.

[a] $p < 0.05$.

[b] $p < 0.05$.

# RESULTS

## Comparison of general data and biochemical parameters between AICS group, TIA group, and healthy control group

There were no significant differences in gender, smoking history, diabetes history and hypertension history between the AICS, TIA, and healthy control groups ($P > 0.05$). The sex ratio (male/female) in the AICS group was 136/112 cases; in the TIA group the ratio was 60/30 cases, and was 102/58 cases in the healthy control group. There was no significant difference between the three groups ($p > 0.05$). The IgE, Lp-PLA2, SAA, LDL-C, TC, TG, and D-D levels in the AICS group were higher than those in the healthy control group and the TIA group, and the differences were statistically significant ($p < 0.05$). The LDL-C and D-D of the TIA group were higher than those of the healthy control group; the difference was statistically significant ($p < 0.05$). IgE was significantly higher in the AICS group than in the healthy control and TIA groups. The results are shown in Table 1.

## Levels of IgE, Lp-PLA2, SAA, and D-D in the AICS plaque group, AICS non-plaque group, and TIA group

According to carotid artery ultrasound examination, the AICS group was divided into the AICS plaque and the AICS non-plaque groups. As shown in Table 2, IgE and Lp-PLA2 in the AICS plaque group were higher than those in the AICS non-plaque group and TIA group. The differences were statistically significant ($p < 0.05$); the IgE level, especially, was significantly increased. There was no difference in SAA and D-D levels between the AICS plaque group and the AICS non-plaque group ($p > 0.05$).

**Table 2** Comparison of IgE, Lp-PLA2, SAA and D-D levels in the AICS plaque group, the AICS non-plaque group and the TIA group.

| index | AICS plaque group (n = 154) | AICS non-plaque group (n = 94) | TIA group (n = 90) | Z value | p value |
|---|---|---|---|---|---|
| IgE (U/ml,M(Q1,Q3)) | 60.00 (34.65,108.68)[ab] | 8.30 (0.00,20.72) | 8.30 (0.00,23.20) | 124.301 | 0.000 |
| Lp-PLA2 (U/L, $\bar{x} \pm s$) | 394.00 (329.00,479.00)[ab] | 343.00 (265.00,422.30) | 257.5 (178.25, 333.00) | 21.665 | 0.000 |
| SAA (mg/L,M(Q1,Q3)) | 5.00 (2.00,11.50)[b] | 4.00 (2.00,14.00) | 3.50 (2.00,4.25) | 0.751 | 0.687 |
| D-D (mg/L,M(Q1,Q3)) | 0.52 (0.21,1.19)[b] | 0.43 (0.21,1.25) | 0.38 (0.18,0.54) | 0.456 | 0.796 |

Notes.
Compared with AICS group.
[a] $p < 0.05$.
Compared with TIA group.
[b] $p < 0.05$.

### Correlation analysis of acute ischemic stroke

Bivariate Spearman rank correlation analysis showed that the incidence of AICS was positively correlated with Lp-PLA2, TC, IgE, TG, D-D, SAA, and LDL-C (r1 = 0.611, 0.499, 0.478, 0.431, 0.386, 0.332, and 0.280, respectively, while all $p < 0.05$). The closer the r value was to 1, the stronger the correlation between this index and the occurrence of AICS disease. The incidence of AICS with plaque was only positively correlated with IgE and Lp-PLA2 (r2 = 0.588, 0.246, $p < 0.05$). Age and sex did not correlate with AICS, and the results are shown in Table 3.

### Logistic regression analysis of influencing factors of AICS

With IgE, Lp-PLA2, SAA, LDL-C, TC, TG, D-D, age and gender as independent variables, and AICS disease with a plaque as dependent variables, multivariate logistic regression analysis showed that IgE and Lp-PLA2 were independent risk factors for predicting the occurrence of AICS disease with plaque ($p < 0.05$). These results were shown in Table 4. The diagnostic value of IgE in AICS with plaque and the sensitivity and specificity of IgE to AICS with plaque are shown in Table 5. The area under the ROC curve (AUC) from large to small was IgE (0.849), Lp-PLA2 (0.646), TG (0.554), LDL-C (0.548), TC (0.545), SAA (0.527), and D-D (0.521). The IgE assay had the highest diagnostic efficacy, sensitivity, and specificity.

## DISCUSSION

Atherosclerotic thrombosis is one of the most significant causes of AICS (*Kounis & Hahalis, 2016*; *Rowland, Hilliard & Barlow, 2015*; *Getz & Reardon, 2019*). The main pathological manifestations are atherosclerotic plaque, which leads to cerebral infarction. The unstable vascular wall plaque can seriously endanger a patient's life, health, and overall quality of life (*Zaha, Joshi & McGuire, 2019*). Dyslipidemia is the leading cause of atherosclerosis (*Li et al., 2010*), and its presence increases the possibility of thrombosis and atherosclerotic plaque formation (*Wiseman et al., 2014*). The stability of carotid atherosclerotic plaques has traditionally been assessed by imaging methods such as ultrasound, CT angiography, or magnetic resonance imaging (*Souza et al., 2020*; *Moss & Ramji, 2016*). However, after the formation of carotid plaques, preventative care and treatments are not as effective. The

**Table 3  Correlation analysis between AICS and multiple indicators of AICS and associated plaques.**

| index | $r_1$ value | $p_1$ value | $r_2$ value | $p_2$ value |
|---|---|---|---|---|
| Age | 0.067 | 0.180 | −0.105 | 0.099 |
| gender | 0.088 | 0.075 | 0.103 | 0.106 |
| LDL-C | 0.280 | 0.000 | 0.095 | 0.113 |
| Lp-PLA2 | 0.611 | 0.000 | 0.246 | 0.000 |
| SAA | 0.332 | 0.000 | 0.046 | 0.474 |
| TC | 0.499 | 0.000 | 0.075 | 0.237 |
| TG | 0.431 | 0.000 | −0.091 | 0.152 |
| IgE | 0.478 | 0.000 | 0.588 | <0.0001 |
| D-D | 0.386 | 0.000 | 0.036 | 0.576 |

Notes.

$r_1$ value, $p_1$ value were correlation analysis results with AICS group, and $r_2$ value, $p_2$ value were correlation analysis results with AICS plaque group.

**Table 4  Logistic regression analysis of AICS plaque prediction indicators.**

| index | $\beta$ value | SE | Wald $\chi^2$ value | OR value (9% CI) | $p$ value |
|---|---|---|---|---|---|
| gender | −0.150 | 0.340 | 0.194 | 0.861(0.443~1.675) | 0. 660 |
| Age | −0.020 | 0.016 | 1.491 | 0.981 (0.950~1.012) | 0.222 |
| Lp-PLA2 | 0.005 | 0.002 | 8.324 | 1.006 (1.002~1.010) | 0.004 |
| IgE | 0.029 | 0.005 | 30.760 | 1.029 (1.019~1.040) | <0.0001 |
| LDL-C | 0.485 | 0.587 | 0.682 | 1.624 (0.514~5.131) | 0.409 |
| SAA | 0.003 | 0.005 | 0.290 | 1.003 (0.993~1.013) | 0.590 |
| TC | −0.631 | 0.550 | 1.315 | 0.532 (0.181~1.565) | 0.251 |
| TG | 0.091 | 0.213 | 0.182 | 1.095 (0.722~1.663) | 0.670 |
| D-D | −0.200 | 0.123 | 2.626 | 0.819 (0.644~1.043) | 0.105 |

Notes.

Age, IgE, Lp-PLA2, SAA, LDL-C, TC, TG and D-D were continuous variables. Gender is referenced to female.

**Table 5  Compares the diagnostic efficacy of AICS with plaque.**

| index | AUC | Youden index | 95% CI | sensitivity (%) | specificity (%) |
|---|---|---|---|---|---|
| IgE | 0.849 | 0.691 | 0. 798~0.891 | 81.82 | 87.23 |
| Lp-PLA2 | 0.646 | 0.290 | 0. 583~0.706 | 66.23 | 62.77 |
| LDL-C | 0.548 | 0.180 | 0.484~0.611 | 35.06 | 83.98 |
| SAA | 0.527 | 0.064 | 0.463~0.590 | 73.38 | 32.98 |
| TC | 0.545 | 0.182 | 0.481~0.608 | 41.56 | 76.60 |
| TG | 0.554 | 0.173 | 0.490~0.617 | 39.61 | 77.66 |
| D-D | 0.521 | 0.974 | 0.457~0.585 | 59.74 | 50.00 |

purpose of this study was to screen out serum or plasma laboratory indicators with high diagnostic value in patients with acute ischemic stroke that was complicated by carotid artery plaque, and to further lay a foundation for establishing laboratory indicators of early warning AICS in the later stage.

LDL-C plays a central role in atherosclerotic heart and brain diseases (*Amarenco et al., 2006*). However, after LDL-C reduction, up to 40% of patients still have the risk of AICS and other cardiovascular and cerebrovascular diseases (*Jang et al., 2021*). Potential laboratory indicators for the early diagnosis and intervention for AICS need to be explored.

A large number of studies have shown that Lp-PLA2, SAA, and IgE may be involved in the occurrence of AS and the formation of vascular plaques (*Liu et al., 2018*; *Shi et al., 2021*; *Fan et al., 2020*; *Diaconu et al., 2021*; *Ozben & Erdogan, 2008*; *Kovanen, 2007*; *Zhang et al., 2020*; *Albers et al., 2002*), while the increase of the D-D level predicts thrombosis in patients (*Albers et al., 2002*).

In an allergic reaction, the immune response shifts to Th2 type (*Lloyd & Hessel, 2010*), and activated Th2 cells can produce cytokines, such as IL-4. These, in turn, may promote the synthesis of allergen-specific IgE by B cells (*Holgate & Polosa, 2008*). IgE is further upregulated and binds to Fc $\varepsilon$R on the surface of the effector cells and activates the synthesis and secretion of a variety of inflammatory mediators by target cells, leading to the continuous progress of inflammation (*Kawakami & Galli, 2002*; *Finkelman, Khodoun & Strait, 2016*). *Wang et al. (2011a)* and *Wang et al. (2011b)* found that serum IgE level in patients with coronary heart disease was significantly increased and correlated with plaque instability. IgE level is also an independent risk factor for diabetes and prediabetes *Wang et al. (2011a)* and *Wang et al. (2011b)*.

This general data statistics of this study showed that the concentration level of IgE, Lp-PLA2, SAA, and D-D in the AICS group were higher than those in the healthy control group and the TIA group. This indicates that the above indicators have diagnostic significance in the laboratory examination of AICS disease. The IgE levels were significantly higher than those in the control group, so its specificity and diagnostic value could be further studied to preliminarily determine whether it is a potential laboratory indicator for early warning of AICS. A comparison of the data between the AICS plaque group and the AICS non-plaque group showed that the levels of IgE and Lp-PLA2 in the AICS plaque group were higher than those in the AICS non-plaque group and the TIA group, and the differences were statistically significant ($p < 0.05$). The IgE level was especially significantly increased. These results confirmed that IgE directly acts on the endothelial cells and the smooth muscle cells, binds to Fc $\varepsilon$R on the surface of effector cells, activates target cells to synthesize and secrete a variety of inflammatory mediators, which directly leads to local inflammatory response and promotes the formation of atherosclerotic plaques. Lp-PLA2 also plays an important role in the formation and development of atherosclerotic plaques. Finally, it plays an important role in the pathogenesis of AICS.

Correlation analysis showed that the incidence of AICS with plaque was only positively correlated with IgE and Lp-PLA2 (r2 = 0.588, 0.246, $p < 0.05$), and the closer the R-value was to 1, the stronger the correlation was with the incidence of AICS. Logistic regression analysis showed that IgE and Lp-PLA2 were independent risk factors for predicting the occurrence of AICS with carotid plaque ($p < 0.05$). The area under the ROC curve (AUC) and sensitivity and specificity data analysis showed that the AUC of IgE (0.849) was significantly higher than that of other indicators, and IgE had the most heightened

sensitivity and specificity. These results indicate that IgE has the highest diagnostic value for AICS with atherosclerotic plaques among these laboratory indicators.

This study revealed that IgE is a potential serologic marker for the diagnosis of acute ischemic stroke with carotid plaques. It can be used as an independent risk factor to predict the occurrence and development of AICS, and can improve the diagnostic level of these diseases for its early prevention, timely diagnosis and treatment.

Since IgE is so valuable for the diagnosis of AICS with atherosclerotic plaques, the further studies can be conducted to determine whether elevated IgE levels occur before the atherosclerotic plaque formation. Whether IgE is a potential laboratory serum indicator for early warning of atherosclerotic plaque formation leading to AICS disease. These need to be further studied.

# CONCLUSIONS

IgE is a serum laboratory indicator used to diagnose AICS disease with carotid plaque, which lays a foundation for further research on potential early warning indicators of AICS disease.

# ACKNOWLEDGEMENTS

We would like to thank Dr. Chongge You for assisting in the revision of the manuscript.

## Funding

This study is supported by the Second Hospital of Lanzhou University Cuiying Science (Grant No.CY2019-BJ14), the Foundation of Education Department of Gansu Province (Grant No.2020b-045), the Second Hospital of Lanzhou University Cuiying Science (Grant No.CY2019-QN20) and Natural Science Foundation of Gansu Province (No.18JR3RA328). The funders had no role in study design, data collection and analysis, decision to publish, or preparation of the manuscript.

## Grant Disclosures

The following grant information was disclosed by the authors:
The Second Hospital of Lanzhou University Cuiying Science: CY2019-BJ14.
The Foundation of Education Department of Gansu Province: 2020b-045.
The Second Hospital of Lanzhou University Cuiying Science: CY2019-QN20.
Natural Science Foundation of Gansu Province: 18JR3RA328.

## Competing Interests

The authors declare there are no competing interests.

## Author Contributions

- Wenwen Jiang conceived and designed the experiments, prepared figures and/or tables, authored or reviewed drafts of the article, and approved the final draft.

![PeerJ]

- Jindou Niu conceived and designed the experiments, performed the experiments, analyzed the data, prepared figures and/or tables, authored or reviewed drafts of the article, and approved the final draft.
- Hongwei Gao performed the experiments, authored or reviewed drafts of the article, and approved the final draft.
- Yingqiang Dang conceived and designed the experiments, performed the experiments, analyzed the data, prepared figures and/or tables, and approved the final draft.
- Meijiao Qi conceived and designed the experiments, analyzed the data, prepared figures and/or tables, and approved the final draft.
- Yumei Liu conceived and designed the experiments, analyzed the data, authored or reviewed drafts of the article, and approved the final draft.

## Human Ethics

The following information was supplied relating to ethical approvals (i.e., approving body and any reference numbers):

The Second Hospital of Lanzhou University granted Ethical approval to carry out the study within its facilities (Ethical Application Ref: 2021A-001).

## Data Availability

The raw data is available as a Supplemental File.

## Supplemental Information

Supplemental information for this article can be found online at http://dx.doi.org/10.7717/peerj.14235#supplemental-information.

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
