# Peer review of "A retrospective study of immunoglobulin E as a biomarker for the diagnosis of acute ischemic stroke with carotid atherosclerotic plaques"

_PeerJ, doi:10.7717/peerj.14235_

## Round 0.1 · original submission · Major Revisions

The reviewers have given their comments and I agree with them as this manuscript requires extensive language and spell-check. Please address the interesting comments raised by the reviewers regarding the discussion which lacks a thorough justification of the level of serum IgE in Stroke patients.

Reviewer 1 ·

Basic reporting

The introduction should cover all the background knowledge of the study. For example, the rationale of the selected biomarkers in ischemic stroke should be described in the introduction.

There are still many misspells and grammatical errors. Some of the errors are shown in annotated PDF.

Experimental design

The definition of the presence or absence of carotid plaque is not clear. Minimal plaque formation is commonly presented in the carotid duplex screening at the age of 60 without any clinical significance. I recommend authors give more details about how they define the carotid plaque. Moreover, the carotid plaque needs to be related to the concurrent stroke (not in the posterior circulation or another side).

The time associated with the onset of stroke for blood sample collection is crucial in this study. However, these important details are lacking.

Validity of the findings

The study may demonstrate the potential use of some biomarkers in acute ischemic stroke. However, this is a retrospective study. I recommend authors add the suggestion of further prospective studies to confirm the use of the biomarkers.

Annotated reviews are not available for download in order to protect the identity of reviewers who chose to remain anonymous.

Reviewer 2 ·

Basic reporting

- The English needs significant improvement throughout the article. The structure of the sentences in this article is confusing, making it difficult to understand its contents. Significant rewriting is needed.
- The background needs to elaborate more on why this testing is needed for early diagnosis. Why is early diagnosis needed? Are there previous studies on the topic? What are the results?
- The discussion repeats the results section. The author should focus on discussing why IgE levels (and other variables) are higher in AICS patients who have developed plaques. Discuss the implication and so on.

Experimental design

- The aim of this study is not so clear to me. What is the meaning of early diagnosis in this article? Will it be used to screen asymptomatic patients so the plaque can be detected earlier? If so, this study was not designed to prove that. Considering that the researchers examine the IgE level after stroke onset, the authors can only say that the IgE helps diagnose AISC with carotid plague. However, they can not say that this is good for early diagnosis. It is not an early diagnosis. The patients already had stroke symptoms and have been examined by CT or MRI.
- The methods are not clear. The authors need to explain in more detail the method they use to measure the variables in the article (IgE, Lp-PLA2, etc.).
- Why only gender and age data were collected? How about history of diabetes, hypertension, etc?

Validity of the findings

As I stated above, the statement in the conclusion that IgE is a serum laboratory indicator of potential early warning of AICS disease with carotid plaque is not supported by the study design and the data found.

Additional comments

No other comments.

---

## Round 0.2 · accepted · Accept

Congratulations your paper is accepted for publication.

Reviewer 2 ·

Basic reporting

No comment

Experimental design

No comment

Validity of the findings

No comment

Additional comments

The authors have addressed all the comments I made before.